# Symbolic Distillation for Learned TCP Congestion Control

**S P Sharan**[1], **Wenqing Zheng**[1], **Kuo-Feng Hsu**[2], **Jiarong Xing**[2], **Ang Chen**[2], **Zhangyang Wang**[1]

[1]University of Texas at Austin    [2]Rice University

{spsharan,w.zheng,atlaswang}@utexas.edu; {kh42,jxing,angchen}@rice.edu

## Abstract

Recent advances in TCP congestion control (CC) have achieved tremendous success with deep reinforcement learning (RL) approaches, which use feedforward neural networks (NN) to learn complex environment conditions and make better decisions. However, such "black-box" policies lack interpretability and reliability, and often, they need to operate outside the traditional TCP datapath due to the use of complex NNs. This paper proposes a novel two-stage solution to achieve the best of both worlds: first to train a deep RL agent, then distill its (over-)parameterized NN policy into white-box, light-weight rules in the form of *symbolic* expressions that are much easier to understand and to implement in constrained environments. At the core of our proposal is a novel **symbolic branching** algorithm that enables the rule to be aware of the context in terms of various network conditions, eventually converting the NN policy into a symbolic tree. The distilled symbolic rules preserve and often improve performance over state-of-the-art NN policies while being faster and simpler than a standard neural network. We validate the performance of our distilled symbolic rules on both simulation and emulation environments. Our code is available at `https://github.com/VITA-Group/SymbolicPCC`.

## 1 Introduction

Congestion control (CC) is fundamental to Transmission Control Protocol (TCP) communication. Congestion occurs when the data volume sent to a network reaches or exceeds its maximal capacity, in which case the network drops excess traffic, and the performance unavoidably declines. CC mitigates this problem by carefully adjusting the data transmission rate based on the inferred network capacities, aiming to send data as fast as possible without creating congestion. For instance, a classic and best-known strategy, Additive-Increase/Multiplicative-Decrease (AIMD) [1], gradually increases the sending rate when there is no congestion but exponentially reduces the rate when the network is congested. It ensures that TCP connections fairly share the network capacity in the converged state.

Figure 1 shows an example where two TCP connections share a link between routers 1 and 2. When the shared link becomes a bottleneck, the CC algorithms running on sources A and B will alter the traffic rate based on the feedback to avoid congestion. Efficient CC algorithms have been the bedrock for network services such as DASH video streaming, VoIP (voice-over-IP), VR/AR games, and IoT (Internet of Things), which ride atop the TCP protocol.

However, it is nontrivial to design a high-performance CC algorithm. Over the years, tens of CC proposals have been made, all with different metrics and strategies to infer and address congestion, and new designs are still emerging even today [2, 3]. There are **two main challenges** when designing a CC algorithm. ①it needs to precisely infer whether the network is congested, and if so, how to adjust the sending rate, based on only *partial or indirect observations*. Note that CC runs on end hosts while congestion happens in the network, so CC algorithms cannot observe congestion directly. Instead, it can only rely on specific signals to infer the network status. For instance, TCP Cubic [4] uses packet loss as a congestion signal, and TCP Vegas [5] opts for delay increase. ②CC algorithms operate within the OS kernel, where the computing and memory resources are limited,

and they need to make real-time decisions to adjust the traffic rates frequently (e.g., per round-trip time). Therefore, the algorithm must be very *efficient*. Spending a long time to compute an action will significantly offset network performance. Over the long history of congestion control, most algorithms are implemented with manually-designed heuristics, including New Reno [6] Vegas [5], Cubic [4], and BBR [2]. In TCP New Reno, for example, the sender doubles the number of transmitted packets every RTT before reaching a predefined threshold, after which it sends one more packet every RTT. If there is a timeout caused by packet loss, it halves the sending rate immediately.

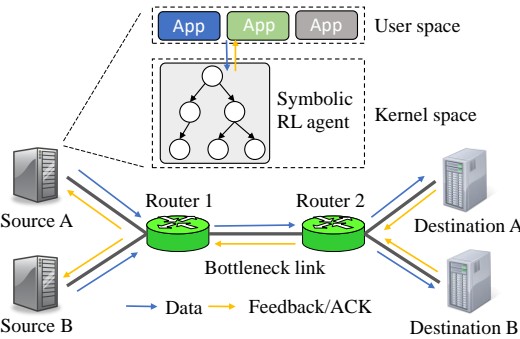

Figure 1: Overview of a congestion control agent's role in the network. Multiple senders and receivers share a single network link controlled by the agent, which dynamically modulates the sending rates conditioned on feedback from receivers.

Unfortunately, manually crafted CCs have been shown to be sub-optimal and cannot support cases that escape the heuristics [7]. For example, packet loss-based CCs like Cubic [4] cannot distinguish packet drops caused by congestion or non-congestion-related events [7]. Researchers have tried to construct CC algorithms with machine learning approaches to address these limitations [7–11]. The insight is that the CC decisions are dependent on traffic patterns and network circumstances, which can be exploited by deep reinforcement learning (RL) to learn a policy for each scenario. The learned policy can perform more flexible and accurate rate adjustments by discovering a mapping from experience, which can adapt to different network conditions and reduce manual tuning efforts.

Most notably, Aurora [7], a deep RL framework for Performance-oriented Congestion Control (PCC), trains a well-established PPO [12] agent to suggest sending rates as actions by observing the network statistics such as latency ratio, send ratio, and sent latency inflation. It achieves competitive results on emulation environments Mininet [13] and Pantheon [14], demonstrating the potential of deep learning approaches to outperform algorithmic, hand-crafted ones. Despite its immense success, Aurora being a neural network based approach, is essentially a black-box to users or, in other words, lacks explicit declarative knowledge [15]. They also require exponentially more computation resources than traditional hand-crafted algorithms such as the widely deployed TCP-CUBIC [4].

## 1.1 Our Contributions

In this work, we develop a new algorithmic framework for performance-oriented congestion control (PCC) agents, which can ①️ run as fast as classical algorithmic methods; ②️ adjust the rate as accurately as data-driven RL methods; and ③️ be simpler than the original neural network [16], potentially improving practitioners' understanding of the model in an actionable manner. We solve this problem by grasping the opportunity enabled by advances in symbolic regression [17–23]. Symbolic regression bridges the gap between the infeasible search directly in the enormous symbolic algorithms space and the differentiable training of over-parameterized and un-interpretable neural networks.

At a high level, one can first train an RL agent through gradient descent, then distill the learned policy to obtain the data-driven optimized yet simpler and easier-to-understand symbolic rules. This results in a set of symbolic rules through data-driven optimization that meets TCP CC's extreme efficiency and reliability demands. However, considering the enormous volume of discrete symbolic space, it is challenging to learn effective symbolic rules from scratch directly. Therefore, in this paper, we adopt a two-stage approach: we first train a deep neural network policy with reinforcement learning mimicking Aurora [7], and then distill the resultant policy into numerical symbolic rules, using symbolic regression (SR).

As the challenge, directly applying symbolic regression out of the box does not yield a sufficiently versatile expression that captures diverse networking conditions. We hence propose a novel branching technique for training and then aggregating a number of SymbolicPCC agents, each of which caters to a subset of the possible network conditions. Specifically, we have multiple agents, each called a branch, and employ a light-weight "branch decider" to choose between the branches during deployment. In order to create the branching conditions we partition the network condition space

into adjacent non-overlapping contexts, then regress symbolic rules in each context. With this modification, we enhance the expressiveness of the resulting SR equation and overcome the bias of traditional SR algorithms to output rules mostly using numerical operators. Our concrete technical contributions are summarized as follows:

- We propose a symbolic distillation framework for TCP congestion control, which improves upon the state-of-the-art RL solutions. Our approach, **SymbolicPCC**, consists of two stages: first training an RL agent and then distilling its policy network into ultra-compact and simple rules in the symbolic expression form.

- We propose a novel branching technique that advances existing symbolic regression techniques for training and aggregating multiple context-dependent symbolic policies, each of which specializes for its own subset of network conditions. A branch decider driven by light-weight classification algorithms determines which symbolic policy to use.

- Through our simulation and emulation experiments, we show that SymbolicPCC achieves highly competitive or even stronger performance results compared to their teacher policy networks while running orders of magnitude faster. The presented model uses a tree structure that is light-weight, and could be simpler for practitioners to reason about and improve manually while narrowing their performance gap.

## 2   Related Works

.   Some proposals use packet loss as a signal for network congestion, e.g., Cubic [4], Reno [24], and NewReno [6]; while others rely on the variation of delay, e.g., Vegas [5], or combine packet loss and delay [25, 26]. Different CC techniques specialized for datacenter networks are also proposed [3, 27].

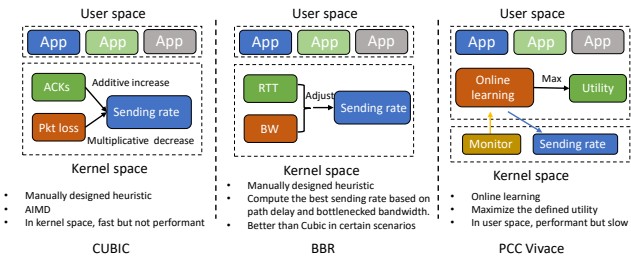

Figure 2: Overview of Conventional Baselines

Researchers have also investigated the use of machine learning to construct better heuristics. Indigo [10] and Remy [11] use offline learning to obtain high-performance CC algorithms. PCC [28] and PCC Vivace [9] opt for online learning to avoid any hardwired mappings between states and actions. Aurora [7] utilizes deep reinforcement learning to obtain a new CC algorithm running in userspace. Orca [8] improves upon Aurora and designs a userspace CC agent that infrequently configures kernel-space CC policies. Our proposal further improves these work.

At the same time, Symbolic regression methods [17–23] have recently emerged for discovering underlying math equations that govern some observed data. Algorithms with such a property are more favorable for real-world deployment as they output white-box rules. [18] use genetic programming based method while [20] uses a recurrent neural network to perform a search in the symbolic space.

We thus propose to synergize such numerical and data-driven approaches using symbolic regression (SR) in the congestion control domain. We use SR by following a post-hoc method of first training an RL algorithm then distilling it into its symbolic rules. Earlier methods that follow a similar procedure do exist, e.g., [29] distills the learned policy as a soft decision tree. They work on visual RL where the image observations are coarsely quantized into $10 \times 10$ cells, and the soft decision tree policy is learned over the 100 dimensional space. [30] also aims to learn abstract rules using a common sense-based approach by modifying Q learning algorithms. Nevertheless, they fail to generalize beyond the specific simple grid worlds they were trained in. [31] learns from boolean feature sets, [32] directly approximates value functions based on a state-transition model, [33] optimizes risk-seeking policy gradients. Other works on abstracting pure symbolic rules from data include attention-based methods [34], visual summary [35], reward decomposition [36], causal models [37], markov chain [38], and case-based expert-behaviour retrieval [21, 22, 39].

## 3   Methodology

Inspired from the idea of "teacher-student knowledge distillation" [40–42], our symbolic distillation technique is two-staged—first train regular RL agents (as the *teacher*), then distill the learned policy

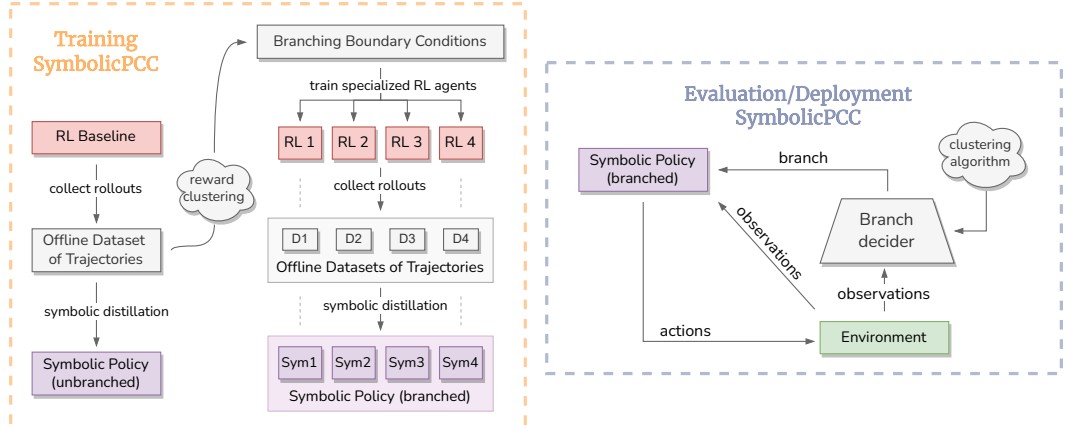

Figure 3: The proposed SymbolicPCC training and evaluation technique: A baseline RL agent is first trained then evaluated numerous times with the roll-outs being saved. Directly distilling out from this data provides a baseline symbolic policy. A light-weight clustering algorithm is used to cluster from the roll-out dataset, non-overlapping subsets of network conditions (aka. branching conditions) that achieve similar return. Separate RL agents are then trained on each of these network contexts and distilled into their respective symbolic rules. During the evaluation, the labels from the clustering algorithm are re-purposed to classify which branch is to be taken given the observation statistics. The chosen symbolic branch is then queried for the action.

networks into their white-box symbolic counterparts (as the *student*). In Section 3.1 we follow [7]'s approach in Aurora and the training of teacher agents on the PCC-RL gym environment. We also briefly discuss the approach of applying symbolic regression to create a light-weight numerical-driven expression that approximates a given teacher's behavior. In Section 3.2 we look at the specifics of symbol spaces and attach internal attributes to aid long-term planning. Finally, in Section 3.3 we discuss our novel branching algorithm as a method for training, then *ensembling* multiple context-dependent symbolic agents during deployment.

### 3.1 Preliminaries: The PCC-RL Environment and the Symbolic Distillation Workflow

PCC-RL [7] is an open-source RL testbed for simulation of congestion control agents based on the popular OpenAI Gym [43] framework. We adopt it as our main playground. It formulates congestion control as a sequential decision making problem. Time is first divided into multiple periods called MIs (monitor intervals), following [28]. At the onset of each MI, the environment provides the agent with the history of statistic vector observations over the network, and the agent responds with adjusted sending rates for the following MI. The sending rate remains fixed during a single MI.

The network statistics provided as observations to the congestion control agent are ① the latency inflation, ② the latency ratio, and ③ the sending ratio. The agent is guided by reward signals based on its ability to react appropriately when detecting changes and trends in the vector statistics of the PCC-RL environment. It is provided with positive return for higher values of throughput (packets/second) while being penalized for higher values of latency (seconds) and loss (ratio of sent vs. acknowledged packets).

**Training of Teacher Agents:**  We first proceed to train RL agents using the PPO algorithm [12] similar to Aurora [7] in the PCC-RL gym environment till convergence. Although they statistically perform very well [7], the PPO agents are entirely black-boxes; this makes it difficult to explain its underlying causal rules directly. Also, their over-parameterized neural network forms incur high latency. Hence, we choose to indirectly learn the symbolic representations using a student-teacher type knowledge distillation approach based on the teacher's (in this case, the RL agent) behaviors.

**Distillation of Student Agents:**  Using the teacher agents, we collect complete trajectories, formally known as roll-outs in RL, in an inference mode—deterministic actions are enforced. The observations and their corresponding teacher actions are MI-aligned and stored as an offline dataset. Note that this step is only performed once at the start of the distillation procedure and is reused in each of its iterative steps. A search space of operators and operands is also initialized (details are discussed

shortly in Section 3.2). Guesses for possible symbolic relations are taken, composed of random operators and operands from their respective spaces. The stored observation trajectories are then re-evaluated based on this rule to output corresponding actions. The cross-entropy loss with respect to the teacher model's actions from the same dataset is used as feedback. This feedback drives the iterative mutation and pruning following a genetic programming technique [44, 45]. The best candidate policies are collected and forwarded to the next stage. If the tree fails to converge or does not reach a specific threshold of acceptance, the procedure is restarted from scratch. Our symbolic distillation method is discussed with further details in our Appendix.

## 3.2 A Symbolic Framework for Congestion Control

**Defining the Symbol Space for CC:** Unlike visual RL [46], the PCC observation space is vector-based, hence we directly plug them into the search space of our numerical-driven symbolic algorithm. We henceforth call these observations as `vector statistic symbols`. The distillation procedure as described earlier learns to *chain* these `vector statistic symbols` using a pre-defined operator space. Specifically, we employ three types of numerical operators. The first type of operators are arithmetic based which include $+, -, *, /, \sin, \cos, \tan, \cot, (\cdot)^2, (\cdot)^3, \sqrt{\cdot}, \exp, \log, |\cdot|$. The second type of operators are Boolean logic, such as $is(x < y)$, $is(x \leq y)$, $is(x == y)$, $a \mid b$, $a$ & $b$, and $\neg a$.

We also utilize a third type of *high level* operators – namely the `slope_of` (observation history) which provides the average slope of an array of observations, and `get_value_at` (observation history, index). The slope operator is especially useful when trying to detect *trends* of a specific statistic vector over the provided monitor interval. For instance, identifying latency increase or decrease trends serves as one of the crucial indicators for adjusting sending rates. Meanwhile the index operator is observed from our experiments to be implicitly used for *immediate responding*—i.e., based on the latest observations.

We note that the underlying decision procedure of the policy network could be efficiently represented in the form of a high-fidelity tree-shaped form similar to Figure 4. This *decision tree* contains said *condition nodes* and *action nodes*. Each condition node forks into two leaf nodes based on the Boolean result of its symbolic composition.

**Attributes for Long Term CC Planning:** In addition to having these operators and operands as part of the symbolic search space, we also attach a few attributes/flags to the agent which are shared across consecutive MI processing steps and help with long-term planning. One behavior in our SymbolicPCC agents is to use this attribute for *remembering* if the agent is in the process of recovering from a network overload or if the network is stable. Indeed, a more straightforward option for such "multi-MI tracking" would be to just provide a longer history of the vector statistics into the searching algorithms. But this quickly becomes infeasible due to an exponential increase of the possible symbolic expressions with respect to the length of `vector statistic symbols`.

## 3.3 Novel Branching Algorithm: Switching between Context-Dependent Policies

Unlike traditional visual RL environments, congestion control is a more demanding task due to the variety of possible network conditions. The behavior of the congestion control agent will improve if its response is conditioned on the specific network context. However, as this context cannot be known by the congestion control agent, the traditional algorithms such as TCP Cubic [4] are forced to react in a slow and passive manner to support both slow-paced and fast-paced conditions. Such a notion of context splitting and branching for training specialized agents can be compared to earlier works in multi-task RL. Specifically, Divide-and-Conquer [47] learns to separate complex tasks into a set of local tasks, each of which can be used to learn a separate policy. Distral [48] proposes optimizing a joint objective function by training a "distilled" policy that captures common behavior across tasks.

Hence, we propose to create $n$ non-overlapping contexts for different network conditions, namely—bandwidth, latency, queue size, and loss rate. We then train $n$ individual RL agents in the PCC Gym by exposing them only to their corresponding network conditions. We thus have a diverse set of teachers which are each highly performant in their individual contexts. Following the same approach as described in Section 3.1, each of the agents are distilled—each one called a *branch*. Finally, during deployment, based on the inference network conditions, the branch with the closest matching boundary conditions/contexts is selected and the corresponding symbolic policy is used.

**Partitioning the Networking Contexts:** A crucial point to note in the proposed branching procedure is to identify the most suitable branching context boundary values. In other words, the best boundary conditions for grouping need to be statistically valid, and plain hand-crafted boundaries are

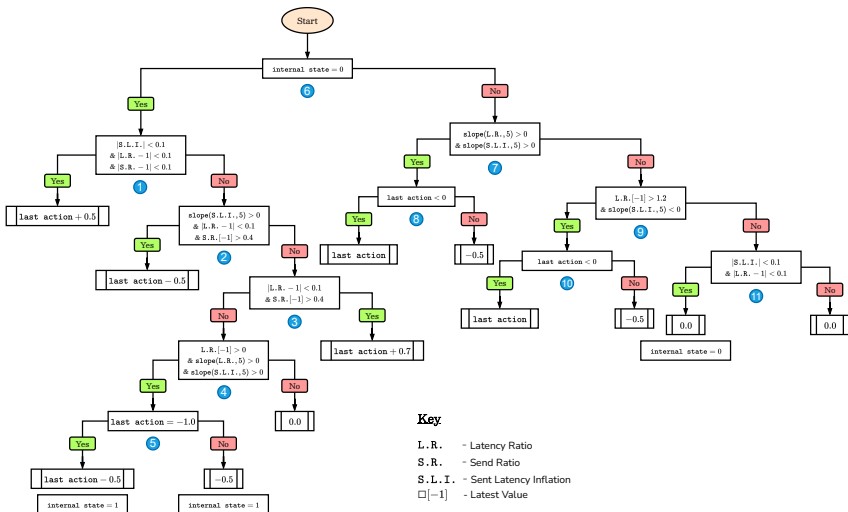

Figure 4: A distilled symbolic policy from the baseline RL Agent in the PCC-RL Environment. Condition nodes are represented as rectangular blocks and action nodes as process blocks.

not optimal. This is because we do not have ground truths of any of the network conditions [49], let alone four of them together. Therefore, we first use a trained a RL agent on the default (maximal) bounds of network conditions (hereinafter called the "baseline" agent). We then evaluate the baseline agent on multiple regularly spaced intervals of bandwidth, latency, queue size, and loss rate and store their corresponding return as well as observation trajectories. To create the optimal groupings, we simply use KMeans [50] to cluster the data based on their return. Due to the inherent proportional relation of difficulty (or in this case the ballpark of return) with respect to a network context, clear boundaries for the branches can be obtained by inspecting the extremes of each network condition within a specific cluster. Our experimentally obtained branching conditions are further discussed in Section 4.2 and Table 1.

**Branch Decider:** Since the network context is not known during deployment, one needs a branch decider module. The branch decider reuses cluster labels from the training stage for a K Nearest Neighbors [51] classification. The light-weight distance-based metric is used to classify the inference-time observation into one of the training groupings and thereby executing the corresponding branch's symbolic policy. Figure 3 illustrates our complete training and deployment techniques.

Lastly, in order to support branching effectively, we have yet another long-term tracking attribute that stores a history of branches taken in order to smooth over any erratic bouncing between branches which are in non-adjacent contexts.

## 4 Experimental Settings and Results

Next, we discuss the abstract rules uncovered by SR, and validate the branching contexts. In Sections 4.3, 4.4, and 4.5 we provide emulation results on Mininet [13], a widely-used network emulator that can emulate a variety of networking conditions. Lastly in Section 4.6, we compare the compute requirements and efficiencies of SymbolicPCC with conventional algorithms, RL-driven methods as well as their pruned and quantized variants. More hyperparameter details are in our Appendix.

### 4.1 Interpreting the Symbolic Policies

The baseline symbolic policy distilled from the baseline RL agent is represented in its decision tree form in Figure 4. One typical CC process presented by the tree is increasing the sending rate until the network starts to "choke" and then balancing around that rate. This process is guided with a series of conditions regarding to inflation and ratio signals, marked with circled numbers in Figure 4. The detailed explanation is in the following.

Condition node ①1 checks whether the `vector statistic symbols` are all stable—namely, whether the latency inflation is close to zero, while latency ratio and send ratio are close to one.

Table 1: The baseline network conditions and resultant branching boundary values (contexts) for each branch after clustering. The return centroid refers to the return value at cluster center of that specific branch.

| Branch | Return Centroid | Bandwidth (pps) | Latency (sec) | Queue Size (packets) | Loss Rate (%) |
|---|---|---|---|---|---|
| Baseline | - | 100 - 500 | 0.05 - 0.5 | 2 - 2981 | 0.00 - 0.05 |
| Branch 1 | 95.84 | 100 - 200 | 0.35 - 0.5 | 2 - 2981 | 0.04 - 0.05 |
| Branch 2 | 576.57 | 200 - 250 | 0.25 - 0.35 | 2 - 2981 | 0.02 - 0.03 |
| Branch 3 | 1046.46 | 250 - 350 | 0.15 - 0.25 | 2 - 2981 | 0.02 - 0.03 |
| Branch 4 | 1516.70 | 350 - 500 | 0.05 - 0.15 | 2 - 2981 | 0.00 - 0.02 |

The sending rate starts to grow if the condition holds. Condition node ②️ identifies if the network is in a over-utilized status `slope_of` (latency inflation) increasing as the key indicator. It the condition is true, the acceleration of sending rate will be reduced appropriately. On the other hand, condition node ③ is activated when the initial sending rate is too low or has been reduced extensively due to ②. ④ is evaluated when major network congestion starts to occur due to increased sending rates from the earlier condition nodes. It checks both latency inflation and latency ratios in an increasing state. Its child nodes start reducing the sending rates and also flip the `internal state` attribute to 1. The latter is used to track if the agent is recovering from network congestion. On the "False" side of ⑥ (i.e. `internal state = 1`), ⑦ and ⑧ realize two stages of recovery, where the latency inflation ratio starts plateauing and then starts reducing. ⑪ indicates that stable conditions have been achieved again and the agent is at an optimal sending rate. The `internal state` is flipped back again to 0 after this recovery.

## 4.2 Inspecting the Branching Conditions

As discussed in Section 3.3, a light-weight clustering algorithm divides the network conditions into multiple non-overlapping subsets. Table 1 summarizes the obtained boundary values. The baseline agent is trained on all possible bandwidth, latency, queue size, and loss rate values, as depicted in the first row. During the evaluation, bandwidth, latency, and loss rate are tested on linearly spaced values of step sizes $50$, $0.1$, and $0.01$, while queue sizes are exponentially spaced by powers of $e^2$ respectively. The return of the saved roll-outs are clustered using K-Means Clustering, and the optimal cluster number is found to be $4$ using the popular elbow curve [52] and silhouette analysis [53] methods. By observing the maximum and minimum of each network condition individually in the $4$ clusters, respective boundary values are obtained. A clear relation discovered is that higher bandwidths and lower latencies are directly related to higher baseline return.

**Remark 1: Exceptions for non-overlapping contexts.** It is also to be noted that no such trend was found between the queue size and return, and hence all the $4$ resultant branches were given the same queue size. A similar exception was made for the loss rates of Branches 2 and 3.

**Remark 2: Interpreting the symbolic policies branches.** All the $4$ distilled symbolic trees from the specialized RL agents possess high structural similarity and share similar governing rules as to that of the baseline agent in Section 4.1. They majorly differ in the numerical thresholds and magnitudes of action nodes, i.e., by varying their "reaction speeds" and "reaction strengths", respectively.

## 4.3 Emulation Performance on Lossy Network Conditions

The ability to differentiate between congestion-induced and random losses is essential to any PCC agent. Figure 5a[1] shows a 25-second trace of throughput on a link where 1% of packets are randomly dropped [54]. As the link's bandwidth is set to 30 Mbps, the ideal congestion control would aim to utilize it fully as depicted by the gray dotted line. Baseline SymbolicPCC shows near-ideal performance with its branched version pushing boundaries further. In contrast, conventional algorithms, especially TCP CUBIC [4], repeatedly reduces its sending rates as a response to the random losses. Quantitative measures of mean square error with respect to the ideal line are provided

---

[1]Interestingly, the figure shows that BBR has rate drop around 11th second. This is a limitation of the BBRv1 design—it reduces sending rate if the `min_rtt` has not been seen in 10s, which is triggered because the RTT in our setup is very stable.

in Table 2 as "Lossy $\overline{\Delta_{opt.}^2}$". This result proves that SymbolicPCC can effectively differentiate between packet loss caused by randomness and real network congestion.

## 4.4 Emulation Performance under Network Dynamics

Unstable network conditions are common in the real world and this test benchmarks the agent's ability to quickly respond to network dynamics. Figure 5b shows our symbolic agent's ability to handle such conditions. The benefits of our novel branching algorithm, as well as switching between agents specializing in their own network context, is clearly visible from faster response speeds. In this case, the link was configured with its bandwidth alternating between 20 Mbps and 40 Mbps every 5 seconds with no loss. Quantitative results from Table 2 show the mean square error with respect to the ideal CC as "Unstable $\overline{\Delta_{opt.}^2}$".

## 4.5 Link Utilization and Network Sensitivities

Link utilization as measured from the server side is defined as the ratio of average throughput over the emulation period to the available bandwidth. A single link is first configured with defaults of 30 Mbps capacity, 30 ms of latency, a 1000-packet queue, and 0% random loss. To measure the sensitivity with respect to a specific condition, it is independently varied while keeping the rest of the conditions constant. An ideal CC preserves high link utilization over the complete range of measurements. From Figure 6, it is observed that our branched SymbolicPCC provides near-capacity link-utilization at most tests and shows improvement over any of the other algorithms.

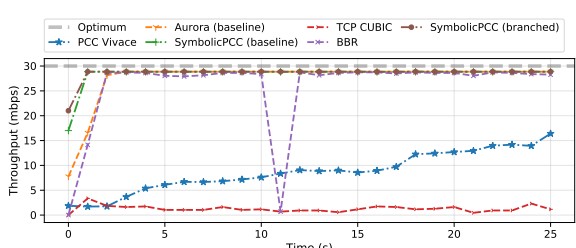

(a) A 25-second thoughput trace for TCP CUBIC, PCC-Vivace, BBR, Aurora, and our SymbolicPCC variants on a 30 Mbps bandwidth link with 2% random loss, 30 ms latency, and a queue size of 1000.

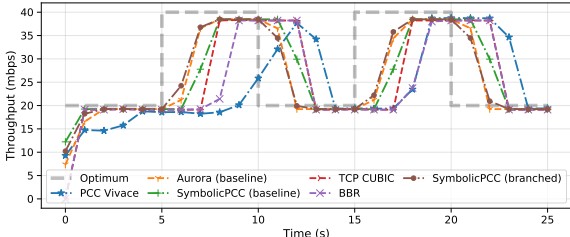

(b) A 25-second throughput trace for TCP CUBIC, PCC Vivace, BBR, Aurora, and our SymbolicPCC variants on a link alternating between 20 and 40 Mbps every 5 seconds with 0% random loss, 30 ms latency, and a queue size of 1000.

Figure 5: Emulation on different conditions.

## 4.6 Efficiency and Speed Comparisons

Since TCP congestion control lies on the fast path, efficient responses are needed from the agents. Due to its GPU compute requirements and slower runtimes, RL-based approaches such as Aurora are constrained in their deployment settings (e.g., userspace decisions). On the other hand, our symbolic policies are entirely composed of numerical operators, making them structurally and computationally minimal. From our results in Table 2, adding the branch decider incurs a slight overhead as compared to the non-branched counterpart. Nevertheless, it is preferable due to its increased versatility in

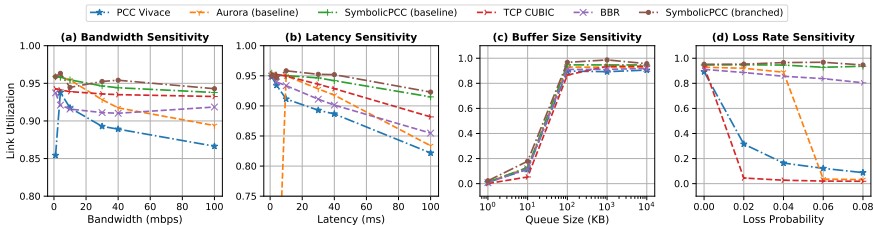

Figure 6: Link-utilization trends as a measure of sensitivities of bandwidth, latency, queue size, and loss rate. Higher values are better.

Table 2: Efficiency and speed comparison of congestion control agents. Note that the ideal values for Lossy $\overline{Thpt.}$ and Lossy $\Sigma Thpt.$ are 30 and 750 respectively (refer Figure 5a).

| Algorithm | Type | FLOPs ($\downarrow$) | Runtime ($\mu s$) ($\downarrow$) | Lossy $\overline{Thpt.}$ ($\uparrow$) | Lossy $\Sigma Thpt.$ ($\uparrow$) | Lossy $\overline{\Delta^2_{opt.}}$ ($\downarrow$) | Oscillating $\overline{\Delta^2_{opt.}}$ ($\downarrow$) |
|---|---|---|---|---|---|---|---|
| TCP CUBIC | Conventional | - | < **10** | 1.27 | 33.01 | 823.02 | 126.07 |
| PCC Vivace | Conventional | - | < 10 | 8.72 | 226.68 | 440.55 | 186.76 |
| BBR | Conventional | - | < 10 | 25.76 | 669.91 | 92.96 | 123.82 |
| Aurora (baseline) | RL-Based | 1488 | 864 | 27.55 | 716.20 | 26.29 | 53.22 |
| Aurora (50% pruned) | RL-Based | 744 | 781 | 27.03 | 709.20 | 27.37 | 61.85 |
| Aurora (80% pruned) | RL-Based | 298 | 769 | 26.42 | 696.86 | 48.13 | 79.80 |
| Aurora (95% pruned) | RL-Based | 74 | 703 | 25.97 | 682.94 | 83.66 | 103.53 |
| Aurora (quantized) | RL-Based | 835 | 810 | 22.54 | 601.78 | 142.92 | 88.45 |
| SymbolicPCC (baseline) | Symbolic | **48** | 23 | 28.40 | 738.46 | 7.29 | 85.03 |
| SymbolicPCC (branched) | Symbolic | 63 | 37 | **28.55** | **742.46** | **4.14** | **43.83** |

different network conditions, as validated by Mininet emulation results. SymbolicPCC achieves **23$\times$ faster execution times** over Aurora, being reasonably comparable to PCC Vivace and TCP CUBIC. We also compare global magnitude pruned and dynamically quantized versions of Aurora. Although these run faster than their baseline versions, they come at the cost of worse CC performance.

## 5 Discussions and Potential Impacts of SymbolicPCC

**Interpretability – a *universal* boon for ML?**   In the PCC domain, the model interpretability is linked to the wealth of domain knowledge. By distilling a black-box neural network into white-box symbolic rules, the resulting rules are easier for the network practitioners to digest and improve.

It may be somewhat surprising that the distilled symbolic policy outperforms Aurora. A natural question arises if it is due to a generalization amplification that sometimes happens for distillation in general or if it is due to symbolic representation. We hypothesize that the performance of a symbolic algorithm boils down to the nature of the environment it is employed in. Congestion control is predominantly rule-based, with deep RL models brought to devise rules more

Table 3: Decoupling: symbolic alone helps generalization.

| Model | Avg. return ($\uparrow$) |
|---|---|
| Aurora | 832 |
| Black-box dist. (50%) from Aurora | 641 |
| White-box dist. from above model | **687** |

complex and robust than hand-crafted ones through iterative interaction. It is only natural to observe that symbolic models outperform such PCC RL models when the distillation is composed of a rich operator space and dedicated policy denoising and pruning stages to boost their robustness and compactness further. To justify this, in Table 3 we analyze the performance obtained by **decoupling distillation and symbolic representation**: we first distill a black-box NN half the size of Aurora ("typical KD") and then further perform symbolic distillation on it.

**On possible limitations.**   We have specifically focused on TCP congestion control as the problem setting,—e.g., the return clustering and reward design. Specific modifications are needed before the approach could be applied to other RL domains.

## 6 Conclusion and Future Work

This work studies the distillation of NN-based deep reinforcement learning agents into symbolic policies for performance-oriented congestion control in TCP. Our branched symbolic framework has better simplicity and efficiency while exhibiting comparable and often improved performance over their black-box teacher counterparts on both simulation and emulation environments. Our results point towards a fresh direction to make congestion control extremely light-weight, via a symbolic design. Our future work aims at more integrated neurosymbolic solutions and faster model-free online training/fine-tuning for performance-oriented congestion control. Exploring the fairness of neurosymbolic congestion control is also an interesting next step. Besides, we also aim to apply symbolic distillation to a wider range of systems and networking problems.

## Acknowledgment

A. Chen and Z. Wang are both in part supported by NSF CCRI-2016727. A. Chen is also supported by NSF CNS-2106751. Z. Wang is also supported by US Army Research Office Young Investigator Award W911NF2010240.

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
