# Appendix: Symbolic Distillation for Learned TCP Congestion Control

**S P Sharan**[1], **Wenqing Zheng**[1], **Kuo-Feng Hsu**[2], **Jiarong Xing**[2], **Ang Chen**[2], **Zhangyang Wang**[1]

[1]University of Texas at Austin    [2]Rice University

{spsharan,w.zheng,atlaswang}@utexas.edu; {kh42,jxing,angchen}@rice.edu

## 1   Algorithm descriptions

We note that the decision procedure of a wide range of policy networks could be efficiently represented as high-fidelity tree shaped symbolic policy. In this tree structure, one basic component – the *condition node*, has three key properties: the *condition*, $a_{LEFT}$, and $a_{RIGHT}$, and could be written equivalent to one basic boolean operation, $condition * a_{LEFT} + \neg condition * a_{RIGHT}$, as explained in Figure 1.

A careful and delicate "DRL behavior dataset" is to be generated and processed, which we specify below. Once having generated the DRL behavior dataset, one could then apply one of the current symbolic regression benchmarks to parse out a symbolic rule that best fit the DRL behavior data.

We now specify how we build the DRL behavior dataset and process into a symbolic regression friendly format. In general, the symbolic regression algorithms are able to evolve into an expression that maps a vector $\mathbf{x} \in \mathbb{R}^d$ into a scalar $y \in \mathbb{R}^1$, where $d$ is the dimensionality of the input vector. To do so, they require a dataset that stacks $N_{\text{Data}}$ samples of $\mathbf{x}$ and $y$, into $\mathbf{X} \in \mathbb{R}^{N_{\text{Data}} \times d}$ and $\mathbf{y} \in \mathbb{R}^{N_{\text{Data}} \times 1}$, respectively. Given these input/output sample pairs, i.e., $(\mathbf{X}, \mathbf{y})$, a symbolic expression that faithfully fit the data can be reliably recovered. The overview of our symbolic distillation algorithm is provided in Table 1 and equivalently in Figure 2.

The genetic mutation is guided by a measure termed program fitness. It is an indicator of the population of genetic programs' performances. The fitness metric driving our evolution is simply the MSE between the predicted action and the "expert" action (teacher model's action). We use the fitness metric to determine the fittest individuals of the population, essentially playing a

---

**Algorithm: Distilling Teacher Behavior into Symbolic Tree**

**Require:**   Temporary dataset $\mathcal{D}_{train}$ containing $\mathbf{X}$ (numerical states), $\mathbf{Y}$ (actions)

**Return:**    $\boldsymbol{r}$: the root of symbolic policy tree

**Maintain:** $\mathcal{S}$: the set of unsolved action nodes

1: **Initializations**
2:     $\boldsymbol{r} \leftarrow newActionNode(depth = 0)$
3:     $\mathcal{S} \leftarrow \{\boldsymbol{r}\}\,; cnt \leftarrow 0$
4: **While** $\mathcal{S} \neq \{\}$ & $cnt < cnt_{MAX}$:
5:     $cnt \leftarrow cnt + 1$
6:     $n \leftarrow pop(\mathcal{S})$    ▷ Sample action node
7:     $\mathbf{Y}_{\text{sub}} \leftarrow \mathbf{Y}[n.\text{total\_condition}]$ ▷ Slices
8:     IF $Entropy(\mathbf{Y}_{\text{sub}}) < \Theta_{entropy}$:
9:         $n.\text{policy} \leftarrow Mean(\mathbf{Y}_{\text{sub}})$
10:   ELSE:        ▷ Single action cannot fit
11:       IF $n.\text{depth} < depth_{MAX}$:
12:           With probability $p_1$: ▷ Split condition
13:               $n \leftarrow newConditionNode()$
14:               $\mathcal{S} \leftarrow \mathcal{S} + \{n.a_{LEFT}, n.a_{RIGHT}\}$
15:           With probability $1 - p_1$:   ▷ De-noise
16:               $n.\text{policy} \leftarrow$ default action
17:       ELSE: ▷ Too deep, stop branching further
18:           With probability $p_2$:
19:               $\mathbf{X}_{\text{sub}} \leftarrow \mathbf{X}[n.\text{total\_condition}]$
20:               $n.\text{policy} \leftarrow runSR(\mathbf{X}_{\text{sub}}, \mathbf{Y}_{\text{sub}})$
21:           With probability $p_3$:       ▷ De-noise
22:               $n.\text{policy} \leftarrow$ default action
23:           With probability $1 - p_2 - p_3$:
24:               $n' \leftarrow Sample(pathToRoot(n))$
25:               $removeSubtree(n')$
26:               $n' \leftarrow newConditionNode()$
27:               $\mathcal{S} \leftarrow \mathcal{S} + \{n'.a_{LEFT}, n'.a_{RIGHT}\}$
28: **Return** $\boldsymbol{r}$

Table 1: Symbolic distillation algorithm.

36th Conference on Neural Information Processing Systems (NeurIPS 2022).

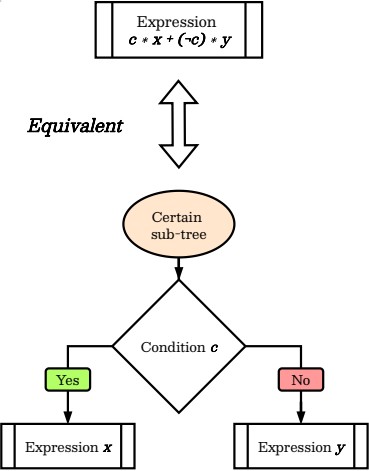

Figure 1: The equivalence of branching node in a subtree and the bool conditioning expression

survival of the fittest game. These individuals are mutated before proceeding to following evolution rounds. We specifically follow 5 different evolution schemes, either one picked stochastically. They are:

- **Crossover:** Requires a parent and a donor from two different evolution tournamets. This scheme replaces (or) inserts a random subtree part of the donor into a random subtree part of the parent. This mutant variant carries forth genetic material from both its sources.

- **Subtree Mutation:** Unlike crossover which brings "intelligent" subtrees into the parent, subtree mutation instead randomly generates it before replacing its parent. This is more aggressive as compared to the crossover counterpart and reintroduce extinct functions and operators into the population to maintain diversity.

- **Hoist Mutation:** Being a bloat-fighting mutation scheme, hoist mutation first selects a subtree. Then a subtree of that subtree is randomly chosen and hoists itself in the place of the original subtree chosen.

- **Point Mutation:** Similar to subtree mutation, point mutation also reintroduces extinct functions and operators into the population to maintain diversity. Random nodes of a tree are selected and replaced with other terminals and operators with the same arity as the chosen one.

- **Reproduction:** An unmodified clone of the winner is directly taken forth for the proceeding rounds.

## 2  Experimental Settings

In our training regime, the configured link bandwidth is between $100 - 500$ pps, latency $50 - 500$ ms, queue size $2 - 2981$ packets, and a loss rate between $0 - 5\%$. In the MiniNet emulation, the link bandwidth is between $0 - 100$ mbps, latency $0 - 1000$ ms, queue size $1 - 10000$ packets, and a loss rate upto $8\%$. The MiniNet configuration is from its default setting, and we adopt this mismatch to purposely explore the model's robustness.

## 3  Extended Discussions

**The Interpretability.**  The simple form of distilled symbolic rules provides more insights for networking researchers of what are the key heuristic for TCP CC. Moreover, our success of using symbolic distillation for CC also paves the possibility of applying it to other systems and networking applications such as traffic classification and CPU scheduling tasks.

```
# psudo-code for the solve stage of RoundTourMix
def solve_policy_as_symbolic_tree(x, y):
    # input is a list of pairs of teacher behaviors:
        # x: numerical state
        # y: action
    # output: a symbolic tree with condition nodes and action nodes
    root = new_action_node(depth=0) # initialize the root node as an action node
    unsolved_action_nodes = { root }
    loop_cnt = 0
    while (unsolved_action_nodes is not empty) and (loop_cnt < max_cnt):
        loop_cnt += 1
        node = sample(unsolved_action_nodes).pop() # randomly sample an unsolved action node
        # First check if the actions under the current total_condition is near deterministic.
        y_subset = y[node.total_condition] # select slices that satisfy total_condition
        if entropy(y_subset) < entropy_threshold:
            # If a single action fits under the current total_condition, then resolve and close this branch
            node.policy = mean(y_subset)
        else:
            if node.depth < max_depth:
                # If max depth is not met, branch on this node by a randomly guessed
                # condition, and mark new child nodes as unsolved
                replace_action_node_with_new_condition_node(node)
                unsolved_action_nodes.add([node.a_LEFT,node.a_RIGHT])
            else:
                # If the current node is already too deep, then stop branching further.
                uniform_0_1 = rand() # sample from a uniform distribtion [0,1]
                if uniform_0_1 > p_SR:
                    # With probability p_SR, directly solve this node using Symbolic_Regression.
                    x_subset = x[node.total_condition]
                    node.policy = Symbolic_Regression(x_subset, y_subset)
                elif uniform_0_1 > p_SR + p_default_action:
                    # With probability p_default_action, set to default action to de-noise teacher behavior.
                    node.policy = default_action
                else:
                    # Otherwise, remove a subtree containing  this node, then renew the searches.
                    node_father = sample(node.father_nodes_list)
                    remove_subtree(node_father)
                    node_father = new_condition_node()
                    unsolved_action_nodes.add([node_father.a_LEFT,node_father.a_RIGHT])
    return root
```

Figure 2: The pseudo-code for the algorithm in Table 1.

**Need for Branching.** The branched training of multiple symbolic models, each in different training regimes, is designed to ease the optimization process. It does **not** directly enforce similarity between solutions for the grouped states – **therefore not causing *brittleness***. This is assured as the symbolic model within any branch does not directly perform the same action for all scenarios within its regime, but contains multiple operations within itself to map states to actions based on the network state observed. Also, during the inference/deployment stage, we use the branch-decider network which chooses branches based on the observed state, **not** the bandwidths or latencies (in fact, these measures are **unavailable** to the controller agent and cannot be observed).