# OpenReview forum: "Symbolic Distillation for Learned TCP Congestion Control"
_NeurIPS.cc/2022/Conference — NeurIPS 2022 Accept_

### Official Review · Reviewer_W18N · 2022-06-19

**Rating:** 6
**Confidence:** 4
**Soundness:** 3 good
**Presentation:** 2 fair
**Contribution:** 3 good

**Summary:**

The authors propose an efficient and interpretable framework for TCP congestion control. Specifically, they first train an RL agent to discover good teacher policies, then distill such policies into students, which are designed to be compact and interpretable using symbolic decision trees. They also split trajectories using Kmeans into contexts, and combine context-specific policies with context regression to further improve performance (essentially using the idea of Multi-task RL).  Extensive experiments demonstrate the effectiveness of the proposed approach.

**Questions:**

1. For trajectory and context clustering phase, why can trajectory returns be clustered in the first place? What is the motivation for it? What semantic meanings do different trajectory returns have?

2. How specifically are rollout trajectories converted into symbolic decision trees (looks like this is based on a standard decision tree process using the rollout actions as input data)? How is tree pruning specifically done? These details should be addressed, using formal mathematics, at least in the supplementary.

3. What is the environment input state space and action space? What semantic meaning does the action space represent? The authors do not study a common setting like vision-based RL, so these should be briefly addressed.

Minor:

"Rewards" should be "Return" in this work. The former means single-step reward, while the latter means episode (aggregated) return.

**Limitations:**

The trajectory clustering part is specific to the environment and reward design authors are using. It is not a general approach that can be applied to any environment. This should be addressed in the limitations.

**Strengths And Weaknesses:**

Strengths: The authors combine RL with symbolic decision tree in the novel application of TCP congestion control. Experiments demonstrate that the proposed approach is very effective and significantly outperforms previous work.

Weaknesses: The paper lacks some formal descriptions of technical details in methodology. Some motivations are not clear and limitations are not addressed. Please see details below.

Overall the methodology does not seem quite novel. The context split and branching process is essentially existing ideas in multi-task RL, and works along this line should be cited [1,2]. However, the application is novel and the approach seems effective, so I still recommend borderline acceptance.

[1] Teh, Yee and Bapst, Victor and Czarnecki, Wojciech M and Quan, John and Kirkpatrick, James and Hadsell, Raia and Heess, Nicolas and Pascanu, Razvan: Distral: Robust multitask reinforcement learning. *Advances in neural information processing systems*, 2017.

[2] Ghosh, Dibya and Singh, Avi and Rajeswaran, Aravind and Kumar, Vikash and Levine, Sergey: Divide-and-conquer reinforcement learning. *International Conference on Learning Representations*, 2018.

**Post Rebuttal**: Thanks for the authors' rebuttal! The added details addressed most of my concerns. Overall, this paper is an application paper that combines existing approaches effectively to a novel domain. While the methodology has limitations, the application itself is novel, and I'm maintaining a stance between borderline and weak accept and increasing my confidence.

---

> ### Author Response · Authors · 2022-08-01
> **Authors' response to reviewer W18N**
>
> **Dear Reviewer W18N:**
>
> We appreciate your carefulness in reviewing our paper. We’ve responded to each of your concerns as follows.
>
> **Weakness #1:** There is a lack of descriptions, such as formal methodological descriptions and two related papers, and ideas borrowed from multi-task RL.
>
> **Reply:**
> Thank you for pointing it out. We have added the necessary descriptions in our Appendix, and have cited the two papers in the new version.
>
> **Question #2:** What is the motivation behind clustering of rewards? What semantic meaning do they convey?
>
> **Reply:** The clustering is necessary because it breaks the problem into simpler subtasks, then develops expert models for each subtask. We see other works use similar approaches as well. For instance, [1] and [2] use delay-loss pairs to cluster data with HMMs into several groups and assign specific sending rates for each group, and [3] clusters the data using into groups using KNNs based on various features based on message size, validity, type, distance, and direction and assign a sending rate for each cluster. On the other hand, it is also increasingly common to find Mixture of Experts in deep learning, performing predictive modeling problems in terms of subtasks using expert models. During inference, individual model weights determined by the gating network are dynamically assigned based on the given input, as the MoE effectively learns which portion of the feature space is learned by each ensemble member implicitly.
>
> In fact, unsupervised clustering-based algorithms for congestion control have been shown to work before where the category of data is unknown, and the sample set needs to be clustered according to the similarity between samples *without carrying explicit semantic meanings*.
>
>
> **Question #3:** What are the observation and action spaces of the environment?
>
> **Reply:** Thank you for your question. The action space is a single continuous float denoting the change of sending rate. Meanwhile, the observation space is a vector of size 3 and contains sent latency inflation, latency ratio, and send ratio. These are explained in Sec 3.1 “Preliminaries”. More technical details regarding the environment are described in Aurora [4].
>
> **Suggestion #4**: “Rewards” should be “return”.
>
> **Reply:** Thank you for the reminder! We are sorry about this typo, and have updated all of them in the newest version.
>
> **Limitation #5:** The trajectory clustering part is specific and not general, which should be addressed in the limitations.
>
> **Reply:**
> We agree that specific modifications are needed before the approach could be applied to other RL domains. We have added new discussion in the limitations section marked in blue.
>
> Given that, this paper is the first one that interprets the learned congestion rules in complex network conditions with rigorous evaluations. Given the practical importance of network congestion control and its complex nature, a grounded and effective interpretation approach is worth its merits.
>
>
> [1] Liu, J., Matta, I., & Crovella, M. (2003, March). End-to-end inference of loss nature in a hybrid wired/wireless environment. In WiOpt'03: Modeling and Optimization in Mobile, Ad Hoc and Wireless Networks (pp. 9-pages).
>
> [2] Barman, D., & Matta, I. (2004, March). Model-based loss inference by tcp over heterogeneous networks. In Proceedings of WiOpt (Vol. 4).
>
> [3] Taherkhani, N., & Pierre, S. (2016). Centralized and localized data congestion control strategy for vehicular ad hoc networks using a machine learning clustering algorithm. IEEE Transactions on Intelligent Transportation Systems, 17(11), 3275-3285.
>
> [4] Jay, N., Rotman, N. H., Godfrey, P., Schapira, M., & Tamar, A. (2018). Internet congestion control via deep reinforcement learning. arXiv preprint arXiv:1810.03259.

---

> > ### Author Response · Authors · 2022-08-06
> > **Kindly expecting your feedback**
> >
> > Dear Reviewer W18N:
> >
> > Since the author-reviewer discussion period has started for a few days, we will appreciate it if you could check our response to your review comments soon. We hope to know if our responses have addressed your concerns on this work, and are sincerely hoping to have further discussions with you.
> >
> > If our responses resolve your concerns, we are kindly expecting an increased rating for our work. Thank you very much for your time and efforts.
> >
> > Best,
> >
> > Authors.

---

### Official Review · Reviewer_aJ6X · 2022-07-09

**Rating:** 5
**Confidence:** 3
**Soundness:** 2 fair
**Presentation:** 3 good
**Contribution:** 2 fair

**Summary:**

This paper presents a novel approach for TCP congestion control, by first learning the  RL policies under different contexts and converting the NN policy into a symbolic policy which consists of several context-dependent policies and switching between the above policies.

**Questions:**

I am curious about the partitioning of the networking contexts.  Did you add any prior that the networking conditions tend to stay the same for a stochastic period? I assume that we can have different networking conditions within one collected trajectory?

**Limitations:**

Although the proposed method is very practical for TCP congestion control, it is of limited novelty. The simple context branching method can not learn good behavior for handling context transitions in the true environment, since there is a delayed consequence of the actions taken in the last context to rewards received in the next context.

**Strengths And Weaknesses:**

Strengths:
1. The method is very practical for TCP congestion control. It retains the performance of RL policy, but is as fast as classical methods and has good interpretability.
2. Networking environments could be very different under different networking conditions/scenarios. For good practical performance, it is necessary to switch between different context-dependent policies.


Weaknesses:
1. The method just applies symbol learning with a context-aware approach to TCP congestion control. The method itself is of limited novelty.
2. In a true environment, the context may transit from one to another. By learning policies under different contexts and then combining them together with the branching method, the learned policy does not know how to deal with the transition between different contexts.
3. New networking scenarios may appear in the true networking environment. The method is not trained in an online fashion, and therefore cannot deal with increasing networking conditions.

---

> ### Author Response · Authors · 2022-08-01
> **Authors' response to reviewer aJ6X - part 2**
>
>
> **Question #4:** Did you add any prior that the networking conditions tend to stay the same for a stochastic period?
>
> **Reply:** We do enforce that within each sample for a specific network condition. We inherit this valid assumption from the Aurora paper, which mentioned that the monitor interval (MI) lasts milliseconds level, and the sample length is set to be no greater than 10. We take 10 samples to define one sample, which still costs ~3 ms level time.
> Although during the following stage of training in the course of which we train specialized agents specific to respective groupings, the network conditions are allowed to stochastically vary although bounded between the extremities of its group. Through these design choices, we believe our method is capable of avoiding high variability and preserving its robustness towards dynamic environmental states.
>
> [1] cite big networking papers that do this to test their congestion controllers
>
> [2] cite big networking names that use mininet for emulation purposes
>
> [3] Wu, F., Yang, W., Sun, M., Ren, J., & Lyu, F. (2020). Multi-path selection and congestion control for ndn: An online learning approach. IEEE Transactions on Network and Service Management, 18(2), 1977-1989.
>
> [4] Lecarpentier, E., Abel, D., Asadi, K., Jinnai, Y., Rachelson, E., & Littman, M. L. (2021, May). Lipschitz lifelong reinforcement learning. In Proceedings of the AAAI Conference on Artificial Intelligence (Vol. 35, No. 9, pp. 8270-8278).
>
> [5] Mendez, J. A., van Seijen, H., & Eaton, E. (2022). Modular lifelong reinforcement learning via neural composition. arXiv preprint arXiv:2207.00429.
>
> [6] Petersen B K, Larma M L, Mundhenk T N, et al. Deep symbolic regression: Recovering mathematical expressions from data via risk-seeking policy gradients[J]. arXiv preprint arXiv:1912.04871, 2019.

---

> > ### Author Response · Authors · 2022-08-06
> > **Kindly expecting your feedback**
> >
> > Dear Reviewer aJ6X:
> >
> > Since the author-reviewer discussion period has started for a few days, we will appreciate it if you could check our response to your review comments soon. We hope our responses have addressed your concerns on this work, and are kindly expecting your feedback.
> >
> > If our responses resolve your concerns, we are kindly expecting an increased rating for our work. Thank you very much for your time and efforts.
> >
> > Best,
> >
> > Authors.

---

> > > ### Comment · Reviewer_aJ6X · 2022-08-08
> > > **Reply**
> > >
> > > Thanks for the authors' detailed response! I believe context is the key concept in this work. Therefore the partition of context, the transition between contexts(and delayed consequence between different policies for sequential contexts), the possible newly encountered context in online learning, and also the generalization between different contexts should be better studied. I will keep my original score due to the practical motivation and performance in TCP congestion control.

---

> ### Author Response · Authors · 2022-08-01
> **Authors' response to reviewer aJ6x - part 1**
>
> **Dear Reviewer aJ6X:**
>
> We highly appreciate your comments and suggestions to make our paper better. We have addressed your concerns below, and have made modifications in the updated paper PDF.
>
> **Weakness #1:** The method just applies symbol learning in congestion control.
>
> **Reply:**
> We respectfully disagree with this statement. First, this paper did not plug any out-of-the-box symbolic regression algorithm. The symbolic regression methods are *column-branching* algorithms, and are *row branching unaware*. However, the proposed method is •column and row branching jointly aware*. Given the data table, previous symbolic regression algorithms yield symbolic expressions that act on the feature columns of the table, in order to match its output to a given column. The resulting symbolic expression could always be written as a tree [6]. However, the proposed algorithm is the first one that jointly branches the row space of the table, to integrate the expert symbolic rules to handle different subtasks.
>
> Second, this paper is the first one that interprets the learned congestion rules in complex network conditions with rigorous evaluations. Given the practical importance of network congestion control and its complex nature, a grounded and effective interpretation approach is worth its merits.
>
>
>
> **Weakness #2:** The learned policy does not know how to deal with the transition between different contexts.
>
> **Reply:** The learned symbolic rule faithfully recovers the RL rule, just using human understandable expressions. The results in table 2 among others have shown that the distilled rule is able to do the task well that its RL surrogate is able to complete. The transition between different tasks is achieved by activations from the branch decider, as discussed in line 245.
>
> Furthermore, the proposed method exhibits performant transitioning behaviors in Fig 5 (b), where the network bottleneck throughput switches continuously, and the proposed method performs the best. Our evaluations are on the popular mininet network emulation testbench which is a very close proxy to real world conditions [2].
>
>
> **Weakness #3:** The method is not trained in an online fashion, therefore cannot deal with increasing networking conditions.
>
> **Reply:** We agree that the method is not trained in an online fashion. However, this is not linked to the inability to handle increasing networking conditions.
>
> First, the proposed method already demonstrates its ability to generalize to unseen conditions: in the evaluation phase, the network condition distribution is wider than training. Specifically, our model was trained on links with bandwidth 100-500 pps, latency 50-500 ms, queue size 2-2981 packets, and a loss rate between 0-5%. In contrast, our mininet emulation system configuration comprises links having bandwidth 0-100 mbps, latency 0-1000 ms, queue size 1-10000 packets, and a loss rate upto 8% to explore our models robustness. Our model does indeed generalize quite well to these emulation conditions and achieves near-perfect link utilization at all capacities as discussed in our results.
>
> Second, previous practitioners in the network congestion control have achieved striking milestones without online learning. Conventional algorithms such as TCP Cubic, or recent RL based SOTA models such as Aurora do not leverage online learning. The results in Fig 5(b) have also not tied the “online learning” to “guaranteed better performance”.
>
> Third, online learning is definitely one of the future scopes, and we see our method can effortlessly be trained in the online fashion just by making the coefficient able to receive gradient. Similar idea has been briefly explored earlier in NDN-MPCC [3] in the PCC domain. Online RL and lifelong learning algorithms are well established and extend traditional RL through recent algorithms such as Lipschitz RMax [4] and accumulated neural compositions [5].

---

### Official Review · Reviewer_Qv2f · 2022-07-10

**Rating:** 5
**Confidence:** 3
**Soundness:** 3 good
**Presentation:** 3 good
**Contribution:** 2 fair

**Summary:**

Reinforcement learning has been previously applied to learn TCP Congestion Control algorithms (in particular, Aurora), but they suffered from not being interpretable, neither could they run in a performant way. The authors propose a two stage framework that builds on top of the approach used in Aurora by distilling the policy through symbolic regression, applying a novel ensembling/branching algorithm that decides the RL agent used. The throughput of their proposed method is compared against existing TCP congestion algorithms (TCP CUBIC, BBR) as well as recent research work (PCC Vivace, Aurora) in varying emulated network conditions and it was found that throughput did indeed improve.

**Questions:**

- Could the authors justify why reward clustering is done rather than clustering based on other network attributes? This may prevent "brittleness" as mentioned in the paper, but I would assume that similar network attributes should be handled similarly, since the same numerical reward may be derived from different network attributes.
- Are you able to briefly characterize the differences for each of the RL agents?
- In Table 2, how does your method compare against BBR?
- Any thoughts on how would you close the gap between runtime for TCP Cubic vs your method?

**Strengths And Weaknesses:**

Strengths
- Good comparisons with other state of the art baselines (CUBIC, Aurora, PCC Vivace) in both runtime and throughput
- Clear diagramming and articulation of technique used
- Interesting albeit simple extension to achieve both performance and interpretability

Weaknesses:
- Lacks theory to explain why the clustering is necessary
- Insufficient reporting of empirical results, e.g. error bars not reported, lack of training details
- lack of proofreading (e.g. line 147 casual instead of causal, Oscillating vs. Unstable for Table 2, min_rrt insteade of min_rtt on pg 7.)

---

> ### Author Response · Authors · 2022-08-01
> **Authors' response to reviewer Qv2f**
>
> **Dear Reviewer Qv2f:**
>
> We highly appreciate your professionality, and your carefulness to raise all these questions. We have replied to all your comments and have made changes to strengthen our paper accordingly.
>
>
> **Weakness & Question #1:** Why is clustering necessary? Why is it based on rewards, and not on other network attributes?
>
> **Reply:**
>
> The clustering is necessary because it breaks the problem into simpler subtasks, then develops expert models for each subtask. We see other works use similar approaches as well. For instance, [1] and [2] use delay-loss pairs to cluster data with HMMs into several groups and assign specific sending rates for each group, and [3] clusters the data using into groups using KNNs based on various features based on message size, validity, type, distance, and direction and assign a sending rate for each cluster. On the other hand, it is also increasingly common to find Mixture of Experts in deep learning, performing predictive modeling problems in terms of subtasks using expert models. During inference, individual model weights determined by the gating network are dynamically assigned based on the given input, as the MoE effectively learns which portion of the feature space is learned by each ensemble member implicitly.
>
>
> Currently we select the rewards as the only attribute for clustering, as we found the rewards are informative enough. We empirically found that the reward acts as the “price” that effectively quantized and labels the complex “costs in the production process” of the network congestion conditions.
>
> **Weakness #2:** Insufficient reporting of training details.
>
> **Reply:**
> Thank you for pointing this out. In the updated paper PDF, more training details are articulated through our main text in blue color as well as in our Appendix section. Also, we do not report error bars as all experiments were only run once.
>
> **Question #3:** What is the difference between RL agents?
>
> **Reply:** In the concern of different RL agents, they all share the same architecture and training schemes - PPO with its default hyperparameters for both our baseline model and specialized agents. There is no difference between the specialized RL agents trained on different groupings of networking regimes apart from the fact that they are targeted towards that specific network condition and exposed to stochastically varying links bounded by those boundaries during training.
>
> **Question #4:** How does the new method compare between BBR?
>
> **Reply:** We have included the corresponding row entry for BBR in our latest revision. Thanks for pointing it out!
>
> **Question #5:** How could we close the runtime gap between TCP Cubic and SymbolicPCC?
>
> **Reply:** We would first like to point out that PCC is a user-space solution whereas TCP Cubic is a kernel-space solution. Since TCP Cubic is integrated at a very low level, i.e. into the linux kernel itself, it is very fast. On the other hand, our proposed method is just a controller, emulating a congestion control in its user-space layer. It could be possible that we move our solution to the kernel in terms of optimized C code and we might achieve similar performances in terms of speed. Our currently presented work only aims towards establishing a simple proof-of-concept implementation while not targeting super efficient implementations. It is very much possible to extend our work for true real world deployments outside controlled testbeds and emulation systems through code optimization and finally integrating it into kernel-space levels, and is the future scope of our work.
>
> **Other concern #6:** There are other typos.
>
> **Reply:** Thank you for pointing these out! We have made the corresponding corrections in our latest submitted revision.
>
> [1] Liu, J., Matta, I., & Crovella, M. (2003, March). End-to-end inference of loss nature in a hybrid wired/wireless environment. In WiOpt'03: Modeling and Optimization in Mobile, Ad Hoc and Wireless Networks (pp. 9-pages).
>
> [2] Barman, D., & Matta, I. (2004, March). Model-based loss inference by tcp over heterogeneous networks. In Proceedings of WiOpt (Vol. 4).
>
> [3] Taherkhani, N., & Pierre, S. (2016). Centralized and localized data congestion control strategy for vehicular ad hoc networks using a machine learning clustering algorithm. IEEE Transactions on Intelligent Transportation Systems, 17(11), 3275-3285.

---

> > ### Author Response · Authors · 2022-08-06
> > **Sincerely expecting further discussions**
> >
> > Dear Reviewer Qv2f:
> >
> > Since the author-reviewer discussion period has started for a few days, we will appreciate it if you could check our response to your review comments soon. This way, if you have further questions and comments, we can still reply before the author-reviewer discussion period ends. If our response resolves your concerns, we kindly ask you to consider raising the rating of our work. Thank you very much for your time and efforts.
> >
> > Best,
> >
> > Authors

---

> > > ### Comment · Reviewer_Qv2f · 2022-08-07
> > > **Reply**
> > >
> > > I'll be sticking to my original rating - the work as other authors have pointed out is an application paper applying existing methods to a novel domain. The choice of return clustering v.s. other possible attributes for clustering should be better studied, and experiments should be repeated so that there's stronger confidence that the results aren't spurious.

---

> > > > ### Author Response · Authors · 2022-08-08
> > > > **Author's response**
> > > >
> > > > We appreciate your feedback.
> > > >
> > > > The novelty of this work should be judged by the amount of new knowledge and new angle it provides to the community and future practitioners, not by whether it was inherited from existing methods or created in place. By distilling the neural network into a symbolic expression, the NN-learned congestion rule are for the first time, converted into a more interpretable expression, which make it easier for the practitioners to analyze and improve. We believe this helps in improving the TCP congestion task magnitudes better than those recently published in neurips-level conferences.
> > > >
> > > > Our experiments are extensively evaluated and repeated in constant throughput, varying throughput, among other settings, with diverse metrics including MSE, loss rate, buffer size, bandwidth, latency sensitiviey, returns, among others. We believe our evaluations are solid.

---

### Official Review · Reviewer_sAC3 · 2022-07-12

**Rating:** 6
**Confidence:** 4
**Soundness:** 2 fair
**Presentation:** 2 fair
**Contribution:** 3 good

**Summary:**

This work deals with the problem of congestion control in TCP communications. Specifically, the work studies policies for TCP senders, which govern the rate at which each sender sends packets.

This problem is a partially observed, multi-agent game as each agent does not have a full picture of the network conditions and instead has to use proxy variables like the presence of a dropped packet to make decisions. Previous solutions have ranged from exploiting manually designed heuristics, to online learning, to deep RL-based methods.

The authors relate that for this problem there is often a tradeoff between agent quality and runtime; while previously developed deep RL agents can well-manage congestion they are too computationally expensive to run in practical settings, which are generally compute constrained.

This work introduces an algorithm claimed to have the best of both worlds, with both a high quality rate determination policy that is also computationally efficient. The proposed method consists of training a DRL agent with PPO that has an accurate rate determination policy, and then distilling the agent into a simple rule-based agent with symbolic regression. The authors train on an OpenAI Gym environment TCP sending setting, and evaluate on mininet.

**Questions:**

Questions are listed in strengths and weaknesses.

**Limitations:**

The authors list Section 5 as the section in which limitations are dealt with but as far as I can tell none are addressed.

**Strengths And Weaknesses:**

This paper presents a performant, computationally-efficient method that could generalize to solving RL problems in other domains. The agent learning algorithm is clear, and the setup is mostly clear. However, the evaluation and motivation sections need further contextualization.

Evaluation: This work requires a more thorough description, and motivation, for the evaluation. Two facets:
- Agent regimes (i.e. kinds of network conditions): What regimes are tested in mininet? Which regimes "trained" on in the RL agent learning environment? Are these regimes multi-agent or single agent? This is important as it is possible for RL agents to "overfit" to a given environment, and to other agents.
- Metrics: Why is the metric squared distance compared to the optimum? A more interpretable value like "throughput" or "regret compared to optimal throughput" seems both more direct and also more human-interpretable.

Motivation: This work lists interpretability as a boon of the algorithm. However, both the exact notion of interpretability and the motivation for wanting interpretability are not fully elucidated.
- It would be good to explain why interpretability is important in this setting, and also what exactly is mean by interpretability [0]. It is said in line 357-358 that interpretability can help researchers "verify the learned model"; however it is unclear how exactly a human would do this. After all, the proposed agent consists of large decision trees (cf. Figure 4); the branching version of the agent is even more difficult to reason about. Decision trees, unless very small and with simple decision points, are difficult for a human to reason about.

[0] https://arxiv.org/abs/1606.03490

---

> ### Author Response · Authors · 2022-08-01
> **Authors' response to reviewer sAc3 - part 2**
>
>
> **Q4:** It would be good to explain the exact notion of interpretability, why interpretability is important, and how interpretability can help researchers.
>
> **Reply**: We refer to the interpretability as *how well a learned controller's behavior can align with the PCC domain knowledge, or be understandable to the network congestion management practitioners*[7]. We stress the need towards practitioners from the general definition, *the ability of the model to explain or to present in understandable terms to a human*[9] , as when it specifically comes to an PCC domain, its interpretability will be uniquely linked to the immense wealth of domain knowledge.
>
> **By distilling a blackbox neural network into a white box symbolic rule, the network congestion practitioners are able to locate the bug, identify the cause, and improve the performance for the learned congestion controller.** The above mentioned domain knowledge could contain important metrics for a network. For example, both our operand and operator space are composed of high level functions which already carry semantic meanings by themselves such as `slope_of` (employed in PCC Vivace [2]), `get_index_of`, ratio of current RTT over minimum connection RTT (latency ratio as employed in Copa [5], and Remy [1]), and the send-ack ratio (employed by BBR [6]). These operators by themselves are good indicators of congestion - for instance unlike a raw latency measurement, which would vary greatly depending on both the base network and the congestion in the network, scale-free metrics like latency ratio should always be around 1.0 if no congestion is present, regardless of the base network and link states. We would like to draw your attention towards Section 4.1 “Interpreting the Symbolic Policies” in our main text where we walk through the regressed symbolic rule in an effortless manner, thanks to our high level operators.
>
> Lastly, TCP CC usually exists in a mixed deployment; as different machines on a network or even across the internet can be found to use different CC algorithms. The ability to analyze how different CC algorithms interact with each other is an important topic. Having interpretability helps with this goal [8].
>
> We appreciate your suggestion, and we’ve added these discussions in Section 5 marked in blue.
>
>
> [1] Winstein, K., & Balakrishnan, H. (2013). Tcp ex machina: Computer-generated congestion control. ACM SIGCOMM Computer Communication Review, 43(4), 123-134.
>
> [2] Dong, M., Meng, T., Zarchy, D., Arslan, E., Gilad, Y., Godfrey, B., & Schapira, M. (2018). {PCC} Vivace:{Online-Learning} Congestion Control. In 15th USENIX Symposium on Networked Systems Design and Implementation (NSDI 18) (pp. 343-356).
>
> [3] Dong, M., Li, Q., Zarchy, D., Godfrey, P. B., & Schapira, M. (2015). {PCC}: Re-architecting congestion control for consistent high performance. In 12th USENIX Symposium on Networked Systems Design and Implementation (NSDI 15) (pp. 395-408).
>
> [4] Tobe, Y., Tamura, Y., Molano, A., Ghosh, S., & Tokuda, H. (2000, November). Achieving moderate fairness for UDP flows by path-status classification. In Proceedings 25th Annual IEEE Conference on Local Computer Networks. LCN 2000 (pp. 252-261). IEEE.
>
> [5] Arun, V., & Balakrishnan, H. (2018). Copa: Practical {Delay-Based} Congestion Control for the Internet. In 15th USENIX Symposium on Networked Systems Design and Implementation (NSDI 18) (pp. 329-342).
>
> [6] Cardwell, N., Cheng, Y., Gunn, C. S., Yeganeh, S. H., & Jacobson, V. (2017). BBR: congestion-based congestion control. Communications of the ACM, 60(2), 58-66.
>
> [7] Doshi-Velez F, Kim B. Towards a rigorous science of interpretable machine learning[J]. arXiv preprint arXiv:1702.08608, 2017.
>
> [8] Ware, R., Mukerjee, M. K., Seshan, S., & Sherry, J. (2019, November). Beyond Jain's Fairness Index: Setting the Bar For The Deployment of Congestion Control Algorithms. In Proceedings of the 18th ACM Workshop on Hot Topics in Networks (pp. 17-24).

---

> > ### Author Response · Authors · 2022-08-06
> > **Sincerely expecting further discussions**
> >
> > Dear Reviewer sAC3:
> >
> > Since the author-reviewer discussion period has started for a few days, we will appreciate it if you could check our response to your review comments soon. We are sincerely expecting further discussions to address any concerns you may have. If our responses resolve your concerns, we are kindly expecting an increased rating for our work. Thank you very much for your time and efforts.
> >
> > Best,
> >
> > Authors.

---

> > ### Comment · Reviewer_sAC3 · 2022-08-07
> > **Response**
> >
> > Re: interpretability
> >
> > "How well a learned controller's behavior can align with the PCC domain knowledge, or be understandable to the network congestion management practitioners" is too abstractly defined to be proved (or be falsified) about a given learned controller. The (very strong) claim that these new models actually aid practitioners ("By distilling a blackbox neural network into a white box symbolic rule, the network congestion practitioners are able to locate the bug, identify the cause, and improve the performance for the learned congestion controller.") needs to be actually shown with an experiment. It is not immediately obvious to this reviewer that the (rather complicated) tree structures that controllers learn in this work are actually (actionably) more interpretable to a practitioner. Trees are not interpretable just because they are made up of simple operations (MLPs also meet this requirement).
> >
> > In general the authors either need to make concrete the claims about interpretability (see: https://arxiv.org/abs/1606.03490) or dial back claims (i.e. "The presented model uses a tree structure that could be simpler for practitioners to reason about and improve manually). Right now the work (see: the title, the contributions, Section 5) claims a much stronger notion of interpretability than is actually shown.
> >
> > This work shows promising directions in both interpretability and reinforcement learning more broadly (not to mention the actual routing problem solved here). Once the concerns around interpretability (and in particular claims about interpretability) are dealt with I will be happy to raise my score.

---

> > > ### Author Response · Authors · 2022-08-08
> > > **Authors' response to reviewer sAc3**
> > >
> > > **Regarding interpretability.**
> > >
> > > **Reply:** We appreciate your suggestion to make this paper more objective. We have changed the title from “Bringing Efficiency and Interpretability to Learned TCP Congestion Control” to “Symbolic Distillation for Learned TCP Congestion Control”, and have modified the contributions and Section 5 accordingly.

---

> > > > ### Comment · Reviewer_sAC3 · 2022-08-08
> > > > **Response**
> > > >
> > > > The misleading claims are still present. For example, in the abstract the authors claim that the presented algorithm is, compared to a DNN, "orders of magnitude faster and interpretable"; the authors also claim on page 2 that the method enjoys "good interpretability" and is "fully interpretable" - neither of these terms are defined in the body. I think the only claim the authors can make here is that the symbolic agent is much simpler than a standard neural network, and that there is some hope that the model is actually more interpretable in a meaningful/actionable sense (that a later paper can show concretely).

---

> > > > > ### Author Response · Authors · 2022-08-08
> > > > > **Author reply**
> > > > >
> > > > > Dear reviewer sAC3:
> > > > >
> > > > > We have just modified the abstract, introduction page 2, the conclusion section, as well as a few other scattered places on the term usage. Please kindly have a check.
> > > > >
> > > > > Best,
> > > > >
> > > > > Authors

---

> > > > > > ### Comment · Reviewer_sAC3 · 2022-08-09
> > > > > > **Response**
> > > > > >
> > > > > > Looks good!

---

> ### Author Response · Authors · 2022-08-01
> **Authors' response to reviewer sAc3 - part 1**
>
> **Dear Reviewer sAc3:**
>
> We genuinely appreciate the weaknesses you have pointed out to strengthen our paper. We have explained all your questions and concerns, and have made modifications to all the weaknesses that you have mentioned. Our replies are detailed below as well as in the updated paper PDF.
>
> **Q1:** What is the difference between training and evaluation regimes? This is important to prevent overfitting the learned congestion controllers.
>
> **Reply**: The training and evaluation in Mininet is slightly different in the network condition. Our model does not overfit to the training regimes, as the evaluation network condition distribution is wider than the training condition.
>
> In the training regime, the link bandwidth is between 100-500 pps, latency 50-500 ms, queue size 2-2981 packets, and a loss rate between 0-5%. In the mininet emulation, the link bandwidth is between 0-100 mbps, latency 0-1000 ms, queue size 1-10000 packets, and a loss rate upto 8%. The mininet configuration is from its default setting, and we adopt this mismatch to purposely explore the model’s robustness. From the results, the model generalizes well to these out bounded emulation conditions. We’ve modified the paper to make this information more clear.
>
> **Q2:** Is it multi-agent?
>
> **Reply:** All our experiments are performed in a single-agent scenario. As we have observed success of symbolic distillation over single agent models, the method shall scale to distilling multi agent models effortlessly, which we set as our future work. We will focus on methods that enforce *fair-play* between an agent and other agents, or between an agent and other legacy congestion controllers to approach global optimization [1], game-theoretic equilibrium analyses [2,3] or path-state classification [4].
>
>
>
> **Q3:** Why is MSE one of our choices of metrics rather than more human-interpretable measures such as throughput?
>
> **Reply:**
> Indeed the throughput is used as our metric, kindly check Figures 5 (a) and (b), where we present our achieved throughputs over a temporal emulation window. The MSE is reported in Table 2, as the “Lossy and Oscillating MSE” metrics. Here, the MSE is exactly used to explain the throughput curves. As explained in line 306, there exists an optimal value, and the MSE here is used to give a numerical estimate of how close different algorithms are compared to that ideal value.
>
> We have presented other human-interpretable metrics too. Figure 6 further included physical significance, namely the link-utilization trends as a measure of sensitivities over a wide range of network conditions. We believe our choice of metrics are appropriate and inclusive from both objective and subjective aspects.

---

> > ### Comment · Reviewer_sAC3 · 2022-08-07
> > **Response**
> >
> > Thank you for the thoughtful response.
> >
> > Re: evaluation and out of distribution/in distribution evaluation:
> >
> > It would be good to separate these categories in the paper so that it is (a) more clear and (b) possible for a reader to understand the performance gap between the two regimes. I also can't find where you made changes wrt this issue in the paper, can you point out where in the paper the changes are?
> >
> > Re: evaluation metrics
> >
> > I meant throughput (maybe average throughput) as a summary metric just as you are using MSE right now (not throughput over time as in Figures 5 and 6). Utilization also seems interpretable (and connected to the actual goal in the setting). What is the advantage of MSE over "average throughput" or "total throughput"?

---

> > > ### Author Response · Authors · 2022-08-08
> > > **Authors' response to reviewer sAc3**
> > >
> > > Thank you for your reply. We address your follow-up questions as follows.
> > >
> > > **Q1:** Where are the out of distribution evaluation settings?
> > >
> > > **Reply:** We have only executed the out of distribution scenario using the default mininet settings in the emulations. The configuration details for the out of distribution experiments are provided in Appendix B.
> > >
> > >
> > > **Q2:** Average and total throughput as evaluation metrics.
> > >
> > > **Reply:** It is reasonable to use average or total throughput instead of MSE of throughput in the “Lossy” case, but not well suited for the “Oscillating” case; avg of an alternating value or the total of a time-sensitive response carry less meaning. Our choice of MSE was to have a consistent metric across both these cases. We have updated our PDF to include both Avg and Total Throughput for the “Lossy” network conditions in Table 2.

---

### Meta-Review · Area_Chair_32Pg · 2022-08-25

**Recommendation:** Accept
**Confidence:** Certain

**Metareview:**

I thank the authors for their submission and active participation in the discussions. The paper presents an RL method for TCP congestion control. While this application paper is borderline, all reviewers unanimously agree that this paper's strengths outweigh its weaknesses. In particular, reviewers remarked that the method is efficient [sAC3], and practical [aJ6X], evaluated well against baselines [Qv2f] with promising results [W18N]. Thus, I am recommending acceptance of the paper but highly encourage the authors to further improve their paper based on the reviewer feedback.

**Award:**

No

---

### Decision · Program_Chairs · 2022-09-14

Accept